# Time-Varying Pattern and Prediction Model for Geopolymer Mortar Performance under Seawater Immersion

**DOI:** 10.3390/ma16031244

**Published:** 2023-02-01

**Authors:** Yingjie Wu, Kun Du, Chengqing Wu, Ming Tao, Rui Zhao

**Affiliations:** 1School of Resources and Safety Engineering, Central South University, Changsha 410083, China; 2School of Civil and Environmental Engineering, University of Technology Sydney, Sydney, NSW 2007, Australia

**Keywords:** seawater immersion, geopolymer, alkali equivalent, waterglass modulus, SVR model

## Abstract

In this study, immersion experiments were conducted on the geopolymer mortar (GPM) by using artificial seawater, and the effects of alkali equivalent (AE) and waterglass modulus (WGM) on the resistance of geopolymer mortar (GPM) to seawater immersion were analyzed. The test subjected 300 specimens to 270 days of artificial seawater immersion and periodic performance tests. Alkali equivalent (AE) (3–15%) and waterglass modulus (WGM) (1.0–1.8) were employed as influencing factors, and the mass loss and uniaxial compressive strength (UCS) were used as the performance evaluation indexes, combined with X-ray diffraction (XRD) and scanning electron microscopy (SEM) to analyze the time-varying pattern of geopolymer mortar (GPM) performance with seawater immersion. The findings demonstrated a general trend of initially growing and then declining in the uniaxial compression strength (UCS) of geopolymer mortar (GPM) under seawater immersion. The resistance of geopolymer mortar (GPM) to seawater immersion decreased with both higher or lower alkali equivalent (AE), and the ideal range of alkali equivalent (AE) was 9–12%. The diffusion layer of the bilayer structure of the waterglass particle became thinner with an increase in waterglass modulus (WGM), which ultimately led to the reduction in the resistance of the geopolymer structure to seawater immersion. Additionally, a support vector regression (SVR) model was developed based on the experimental data to predict the uniaxial compression strength (UCS) of GPM under seawater immersion. The model performed better and was able to achieve accurate prediction within 1–2 months, and provided an accurate approach to predicting the strength of geopolymer materials in a practical offshore construction project.

## 1. Introduction

The durability of concrete is an important basis for assessing its performance in an aggressive environment. It is known that concrete structural degradation and a considerable loss in service life are caused by seawater erosion in marine engineering. The more reliable explanation for why objects deteriorate is because seawater is high in corrosive ions such as Cl^−^, SO_4_^2−^, and Mg^2+^, which react with Ca(OH)_2_ and hydrated calcium aluminate to produce insoluble salts in weakly alkaline environments [1]. Insoluble salts tend to produce large volume expansion followed by the formation of cracks, and the further dissolution of ions leads to the deepening of cracks, eventually leading to a loss of concrete structure. The geopolymer material can effectively solve this problem because of its excellent erosion resistance and mechanical properties. In addition, the production of geopolymer material is environmentally friendly and studies have shown that the production of fly-ash-based geopolymer has lower CO_2_ emissions, at least 11% lower than Portland cement [2]. It has become a new cement replacement material.

Due to its excellent erosion resistance and mechanical properties, the geopolymer cementitious material has become a friendly substitute to solve the problem [3,4,5]. Owing to the significant environmental advantages, geopolymer materials have become a hot topic of research in recent years, with performance-influencing factors receiving a lot of attention from researchers [6,7]. The aggregate type, activator concentration, activator type, activator ratio, etc., are the primary variables determining the performance of geopolymer materials. Wang et al. [8] investigated the effect of NaOH solution concentration on the mechanical and chemical properties of metakaolin-based polymers and found that the flexural strength, compressive strength, and apparent density of the metakaolin-based polymers increased with increasing NaOH solution concentration. Qiu et al. [9] revealed that high OH^-^ concentrations in the activator caused cementitious building materials to lose strength. Dai et al. [10] investigated the effect of activator type on the properties of geopolymers by preparing geopolymers with two different activators, NaOH and KOH. The results showed that the KOH-based geopolymer had better mechanical properties. Ground granulated blast furnace slag (GGBS)/fly ash ratio and water-binder (W/B) ratio are also important factors affecting the performance of geopolymer materials. GGBS/fly ash ratio is the main factor, while the effect of W/B ratio depends on the ratio of GGBS/fly ash [11]. Recent years have seen a significant increase in research into the erosion resistance of geopolymers, particularly with regard to seawater corrosion resistance [12,13,14,15]. Xiao et al. [16] effectively developed high-performance marine concrete using a dual admixture of mineral admixtures and anti-corrosion additives after analyzing the dissolution process of ions in seawater on concrete using artificial seawater simulation testing. Chindaprasirt et al. [17] investigated the effect of NaOH concentration in a marine environment on chloride penetration and the compressive strength of geopolymer concretes, and the results showed that a high concentration of NaOH helps to enhance the compressive strength and reduce the chloride penetration of geopolymer concrete. Ouda et al. [18] focused on the thermal stability and resistance to seawater immersion of fly ash geopolymers doped with various percentages of tile waste compared to normal blends, and showed that the addition of 10% tile waste enhanced the compressive strength and microstructural properties both before and after exposure to seawater solution. Dai et al. [10] investigated the effect of activator type on the properties of geopolymers by preparing geopolymers with two different activators, NaOH and KOH. The results showed that the KOH-based geopolymer had better mechanical properties.

For geopolymer performance prediction, Behnood et al. [19] used the M5P algorithm to examine high-performance-concrete compressive strength prediction by building model trees from high-dimensional data. In order to solve civil engineering issues more accurately, Chou et al. [20] created a support vector regression system based on the artificial intelligence firefly swarm algorithm. In order to increase the precision of high-performance-concrete compressive strength prediction by parameter optimization, Omran et al. [21] investigated the impact of three different materials (fly ash, Haydite lightweight aggregate, and Portland limestone cement) on the compressive strength of high-performance concrete using the bagging integration algorithm. However, few related scholars have focused on the prediction of the time-varying performance of geopolymers under seawater immersion.

Numerous researchers have conducted some studies on the durability performance of geopolymer materials under seawater immersion and have demonstrated that geopolymer materials have excellent resistance to seawater immersion. However, these studies only focus on the relationship between the influencing factors and durability, and do not systematically analyze the time-varying patterns of geopolymer materials with seawater immersion. In addition, a large number of studies are only based on experimental results without subsequent performance prediction. Therefore, this study focused on analyzing the influence law of activator parameters and the time-varying law of performance under seawater immersion conditions, and establishing a prediction model for performance prediction by combining the experimental results.

This study examined the effects of waterglass modulus (WGM) and alkali equivalent (AE) on the mechanical characteristics and durability of geopolymer mortar (GPM) under seawater immersion utilizing GPM specimens subjected to artificial seawater immersion for 270 days. AE (3%, 6%, 9%, 12%, and 15%) and WGM (1.0, 1.2, 1.4, 1.6, and 1.8) were used as influencing factors, and mass loss and uniaxial compressive strength (UCS) were used as performance evaluation indexes, combined with X-ray diffraction (XRD) and scanning electron microscopy (SEM) to analyze the time-varying patterns and intrinsic causes of GPM performance with seawater immersion. To further realize the prediction of the strength of GPM under seawater immersion, a support vector regression (SVR) model was developed based on the experimental findings, which provides a basis and reference for the application of geopolymer materials in actual offshore construction projects.

## 2. Experimental Program

### 2.1. Raw Materials

Fly ash (FA), silica fume (SF), and slag were employed as three separate raw materials in this study to cast GPM. The slag is S95 ground granulated blast furnace slag in white powder form and the fly ash is low-calcium (CaO < 10%), and Table 1 details the chemical makeup of each. SiO_2_ makes up 92% of the mass of SF, with an average particle size of 0.1 μm, and the specific properties of the materials are shown in Table 2. Natural river sand with a medium fineness makes up the fine aggregate. As an alkaline activator, sodium silicate solution and NaOH were chosen. NaOH is a 99% pure flake, and the sodium silicate solution has a modulus of 3.35 (8.6% Na_2_O, 27.9% SiO_2_, and 63.5% H_2_O).

### 2.2. Sample Preparation

*Ae* and *Mo* were used as variables in GPM specimens to study the impact of AE and WGM on the performance of GPM under seawater immersion circumstances. The parameters are defined in Equations (1) and (2). *Ae* ranged from 3 to 15%, *Mo* ranged from 1.0 to 1.8, and the mass ratio of sand aggregate to base material was 2.0. The specimens were split into two groups of A and M for preparation. Table 3 and Table 4 show the precise matching ratio.

*Ae* is the mass ratio of the equivalent basic oxide (Na_2_O) in the activator (Na_2_SiO_3_) to the activated solid powder, and it is computed as follows:(1)$$Ae=\frac{m({\mathrm{Na}}_{2}O)}{m(\mathrm{Raw}\;\mathrm{material}\;\mathrm{powder})}\times 100\%$$
*Mo* is the molar ratio of equivalent SiO_2_ to equivalent basic oxide (Na_2_O) in the activator (Na_2_SiO_3_), and it is computed as follows:(2)$$Mo=\frac{mol({\mathrm{SiO}}_{2})}{mol({\mathrm{Na}}_{2}O)}\times 100\%$$

The alkaline activator was prepared by sodium hydroxide flakes and sodium silicate solution 24 h in advance. To lessen the impact of unknown impurities, sodium hydroxide flakes were dissolved in distilled water, mixed proportionately with sodium silicate solution, and kept at room temperature (23 ± 2 °C, 65% relative humidity) for storage. The base material and sand aggregate were mixed in the mixer for 20 min and cooled to room temperature. Then, the prepared alkali activator was added and stirred into shape. A normal mortar cube of 70.7 mm in height, width, and depth was used as the specimen, with 30 specimens in each group. The surface of the specimen was covered with plastic film after it was manufactured to stop moisture loss and cracking. The samples were exposed to a standard curing chamber temperature of 23 °C ± 2 °C for 24 h before being demolded and tagged. The specimen was kept in the maintenance room with film and water after it was demolded until the 28 days test age.

### 2.3. Test Method

The maintained GPM specimens were immersed in artificial seawater. The artificial seawater liquid level was at least 50 mm above the specimen and the specimen interval was not less than 50 mm. Plastic film was used to seal the immersion chamber and the artificial seawater was replaced every 30 days. The mass change test was conducted for fixed specimens throughout the seawater immersion phase in order to diminish the impact of individual specimen variances on the test findings. The samples were periodically removed from the immersion chamber and put into a drying chamber at a constant temperature of 105 °C for drying until the quality no longer changed. The samples were weighed using an electronic scale with an accuracy of 0.01 g. A WES-600 universal testing machine was used to determine the UCS of the samples. The fine morphological characterization of GPM was performed by SEM using JEOL’s JSM-IT500 (Akishima, Japan), and the reaction products and compositions were analyzed by XRD. The artificial seawater was prepared with reference to the standard ASTM D 1141-98 (2013), and the main salt contents of the artificial seawater are shown in Table 5.

The configuration of artificial seawater with ion concentration was increased to three times in order to speed up immersion and reduce the test period. The main dissolved ions and their mass percentages are presented in Figure 1, where the cations are primarily Na^+^ and Mg^2+^ while the anions are primarily Cl^−^ and SO_4_^2−^. Artificial seawater was prepared and replaced monthly to weaken the effect of water evaporation on seawater concentration.

### 2.4. UCS Prediction Model of GPM

It is evident that while the geopolymer material has good resistance to seawater immersion, structural degradation with increasing seawater immersion time is unavoidable. In actual offshore construction projects, if the strength of the geopolymer structure can be predicted in advance, engineering disasters caused by structural deterioration can be effectively avoided. In order to give an effective and scientific technique for the precise prediction of GPM strength under seawater immersion, this research regressed and predicted the time-varying UCS of GPM under seawater immersion based on the support vector regression (SVR) model.

#### 2.4.1. Support Vector Regression (SVR)

Support vector machine (SVM) is a better algorithm for solving small-sample, nonlinear, and high-dimensional problems based on the statistical Vapnik–Chervonenkis (VC) dimensional theory and the principle of structural risk minimization [22]. SVR is an important branch of SVM for solving regression problems, which is to find an optimal classification surface by training set samples so that the error of all samples from that classification surface is minimized [23]. The prediction of the test set was performed using the regression function. The mean square error *MSE* and the coefficient of determination *R*^2^ of the test set were recorded and calculated as:(3)$$MSE=\frac{1}{l}\sum_{i=1}^{l}{\left({\hat{y}}_{i}-y_{i}\right)}^{2}$$
(4)$$R^{2}=\frac{{\left(l\sum_{i=1}^{l}{\hat{y}}_{i}y_{i}-\sum_{i=1}^{l}{\hat{y}}_{i}\sum_{i=1}^{l}y_{i}\right)}^{2}}{\left(l\sum_{i=1}^{l}{\hat{y}}_{i}{}^{2}-{\left(\sum_{i=1}^{l}{\hat{y}}_{i}\right)}^{2}\right)\left(l\sum_{i=1}^{l}y_{i}{}^{2}-{\left(\sum_{i=1}^{l}y_{i}\right)}^{2}\right)}$$
where *l* is the number of samples in the test set; *y_i_* (*i* = 1, 2, …, *l*) is the true value of the *i*th sample; ${\hat{y}}_{i}$(*i* = 1, 2, …, *l*) is the predicted value of the *i*th sample.

According to *MSE* and *R*^2^, the established regression models were assessed. Changing the kernel function and other model parameters until the performance satisfies the requirements was performed to the model if the performance did not satisfy the requirements.

#### 2.4.2. The SVR Prediction Model

The SVR prediction model only considers the effects of three factors, AE, WGM, and seawater immersion time. In order to better evaluate their effects on the compressive strength of GPM, the data of A and M groups were processed independently. Let the sample be {(*x_i_*, *y_i_*), *i* = 1, 2, …, 50}, where *x_i_* is the influence factor of the *i*th sample, and *y_i_* is the UCS corresponding to the *i*th sample. The measured data (No. 1–45) before 270 days of immersion were used as the training set, and the measured data (No. 46–50) at 270 days of immersion were used as the test set to construct the SVR prediction model. The sample data are shown in Table 6 and Table 7.

In the analysis of GPM performance prediction under seawater immersion, it was found that the Radial Basis Function (RBF)$K(x_{i},x)=\exp(-{\left\|x_{i}-x\right\|}^{2}/2\varepsilon^{2})$ has high computational accuracy [24,25]. Consequently, considering the model prediction accuracy and generalization ability, RBF was selected as the kernel function, and cross-validation and grid search were implemented to obtain the best penalty factor *C* and kernel function variance *g*. The insensitivity coefficient *ε* = 0.01 was set to generate the SVR model.

## 3. Results and Discussion

### 3.1. Mass Change

Table 8 and Table 9 display the mass changes of the GPM specimens under seawater immersion conditions.

The GPM specimens displayed an overall trend of growing and subsequently reducing mass change during the seawater immersion phase, as shown in Table 8 and Table 9. This indicates that a higher rate of GPM hydration products was produced than hydrolysis during the early stages of seawater immersion. The specific cause is that the tetrahedral A1O_4_^−^ in the geopolymer mesh structure has negative charge and needs cations such as Na^+^ and Mg^2+^ to balance the charge [26]. As a result, high concentrations of Na^+^ and Mg^2+^ in the seawater prior to immersion diffuse into the interior of the geopolymer, resulting in an increase in mass. Notably, sea salt crystals were discovered during the SEM investigation (Figure 2d), which may also lead to a change in the mass of the GPM. In the late stages of immersion, the rate of hydration product formation declined at the same time as the rate of internal gel phase hydrolysis increased, leading to the phenomena of mass reduction, i.e., the release of aluminate and silica into solution from the gel structure [27]. Table 8 responds to the effect of AE on the change in GPM mass, and it is clear that the mass reduction in GPM happened at immersion in 180 days when AE was between 3% and 9%. The mass reduction phenomenon was advanced to 120 days when AE was further enhanced. The reason could be that the excess sodium remained in the sample and weakened the structure of the geopolymer, which triggered an early hydrolysis of the gel phase structure [28]. Table 9 provides an illustration of the impact of WGM on the mass change of GPM, and it is clear from Table 9 that WGM had no effect on the trend of its mass growth throughout the initial stage. When WGM was between 1.0 and 1.6, the rate of mass change fluctuated around 1% without an obvious pattern and no additional mass reduction occurred with the increase in seawater immersion time (>60 days). This suggests that the silica-alumina mesh structure of GPM was not harmed, showing that seawater immersion had less of an impact on GPM. Analysis of the cause for the lack of additional mass reduction suggests that it may be related to the low solubility of silica, which prevents further erosive action of seawater when it precipitates on the contact surface between GPM and seawater [29]. Additionally, the unreacted silicate particles in geopolymer acted as insoluble fillers, enhancing their chemical resistance. After 270 days of seawater immersion, none of the test groups revealed significant mass loss, indicating that the GPM structure had good resistance to seawater immersion and had not suffered significant damage.

### 3.2. Time-Varying Law of Compressive Strength

#### 3.2.1. Initial Compressive Strength

The initial uniaxial compressive strength of GPM was measured when they were not eroded by seawater, and the results are shown in Figure 3. The correlation between AE, initial compressive strength, and rate of strength change is illustrated in Figure 3a. The initial compressive strength of GPM showed a trend of increasing and then decreasing with the increase in AE. In the case of AE less than 9%, the compressive strength increased with AE, and when it was more than 12%, the compressive strength showed a decreasing trend. At AE of 9%, the compressive strength reached its maximum value of 74.19 MPa. The compressive strength was only 41.08 MPa at AE of 3%, which is because the lower AE failed to allow the aggregates to react adequately, and a significant portion of unreacted aggregates were wrapped by the cementitious material and eventually formed a relatively loose gel phase, reducing its strength. This conclusion was supported by the corresponding SEM image (Figure 2a). When AE gradually increased to 12%, the compressive strength of GPM continuously increased while the strength growth rate kept decreasing, and the strength growth rate of AE from 9% to 12% was almost 0. The compressive strength decreased when AE increased from 12% to 15%, which could be caused by the high alkali concentration. Our results are in line with those of Alonso et al. [30], demonstrating that a high concentration of NaOH inhibits the synthesis of geopolymers. The link between WGM and GPM compressive strength and the rate of change of strength is shown in Figure 3b. As seen in Figure 3b, WGM increased from 1.0 to 1.8, whereas the corresponding strength reduced from 76.95 MPa to 68.54 MPa, a 10.9% loss in strength. It can be seen that although the UCS of GPM fluctuated slightly with the increase in WGM, the overall trend was downward and there was an inverse correlation. There was no discernible trend in the rate of change in its strength, which varied between −6.46% and 0.35%. The decrease in GPM strength was because the elevation of WGM leads to the thinning of the diffusion layer in the waterglass structure, the transfer of Na^+^ and OH^-^ to the compact layer, and the decrease in free Na^+^ and OH^-^ content. This leads to the inability of Ca^2+^, Si^2+^, and Al^3+^ in the aggregate to be fully activated, which eventually leads to the decrease in the strength of the formed condensed structure [31]. It is worth noting that too low a WGM (*Mo* < 0.8) during the test causes the solution to quickly crystallize, preventing it from effectively participating in the polycondensation reaction to generate strength.

#### 3.2.2. The Impact of AE

Figure 4a illustrates the correlation between the UCS of GPM specimens, seawater immersion time, and AE during simulated seawater immersion. From the upper surface of Figure 4a, it is clear that the UCS of GPM showed a trend of increasing and then decreasing with both immersion time and AE. From the bottom projection of Figure 4a, it can be seen that the optimum strength range of the GPM specimens occurred between 9% and 12% for AE and 30 d and 90 d for immersion time, with a maximum value of 78.73 MPa at 9% for AE and 30 d for immersion time. In each stage of immersion, the trend of UCS with AE showed an increasing trend followed by a decreasing trend, which was the same as the initial strength change pattern without seawater immersion. When AE was 3–12%, the UCS of GPM grew rapidly with immersion time in the initial stage of immersion (<60 days), then declined as immersion time increased (60–90 days). The rate of decline in UCS slowed down and stabilized when the structure became stable (90–270 days). This indicates that short-term seawater immersion can increase the UCS of GPM. The UCS of GPM gradually decreased as immersion time increased, which is consistent with Ouda’s findings [18]. The large amounts of metal cations such as Mg^2+^ and K^+^ in seawater that enter the reticular structure of GPM and replace the Na^+^ equilibrium charge to stabilize the silica-aluminate reticulation are to blame for the strength enhancement in the initial stage. This is consistent with the increment in the mass of GPM specimens in the pre-seawater immersion period. Contrarily, the UCS of GPM decreased at a later stage with increasing immersion time, which was attributed to a decrease in calcium content [32]. Chi and Huang [33] attributed this to the decreased activity of fly ash. Combined with the XRD pattern (Figure 5) and SEM images (Figure 2), it was discovered that no appreciable amount of destructive material was created. Therefore, this loss in UCS was more likely caused by the weakening of the gel phase structure.

Figure 6a shows the strength loss rate of GPM specimens after 270 days of seawater immersion. As seen in Figure 6a, when AE was 3%, the compressive strength loss rate was lowest at 4.31%. Its compressive strength loss rate varied from about 10% when AE was between 6% and 12%. When AE was 15%, the UCS of GPM dropped from 67.41 MPa to 52.44 MPa, a reduction of 22.21% in strength. It is noteworthy that the strength enhancement phenomenon of pre-seawater immersion did not occur when AE was 15% (Figure 4a), which shows that higher AE leads to a decrease in compressive strength of GPM and a decrease in resistance to seawater immersion. This finding that a higher AE generates more free bases and free bases result in increased detrimental qualities such as brittleness is also supported by Wang et al. [34]. They concluded that the optimal range of AE was from 3% to 5.5%, which was smaller than the optimal range from 9% to 12% in this test. This decrease in the optimal range is attributable to the thorough consideration of performance and economy.

#### 3.2.3. The Impact of WGM

The change pattern of the UCS of GPM specimens with seawater immersion time and WGM is shown in Figure 4b. It is easy to see from the upper surface of Figure 4b that the immersion time of 60 days displayed a raised shape, indicating that the change in WGM did not affect the phenomena of the increment in compressive strength in the initial stage of seawater immersion. Unlike the pattern of compressive strength change brought on by AE, the strong compressive strength surface was “downhill-shaped” with an increase in WGM, meaning that following seawater immersion, the UCS of GPM dropped with an increase in WGM. According to the compressive strength projection at the bottom of Figure 4b, the optimum strength range of GPM specimens occurred when WGM was 1.0–1.4 and the immersion time was 30 days~90 days. When WGM was 1.0 and immersion time was 30 days, the UCS of GPM attained a maximum value of 81.18 MPa, and a minimum value of 54.83 MPa when WGM was 1.8 and the immersion period was 270 days. It can be seen that the increase in WGM and silica content resulted in a decline in compressive strength rather than an increase in strength. The link between compressive strength loss rate and WGM was plotted (Figure 6b) in order to further study the relationship between WGM and the compressive strength of GPM following seawater immersion. As demonstrated in Figure 6b, the compressive strength loss rate varied between 10 and 15% when GWM was less than 1.6, indicating that the structure had not been seriously harmed. When GWM increased to 1.8, the UCS of GPM decreased from 68.541 MPa to 54.83 MPa, a strength loss of around 20%, and the resistance to seawater immersion sharply declined, demonstrating that the increase in WGM resulted in a reduction in the resistance of GPM to seawater immersion. Wang et al. [34] similarly found that the strength of alkali-activated slag diminished with increasing modulus after WGM was larger than 1.5. Xie et al. [35] found that an increase in NaOH content decreased the modulus of the sodium silicate solution and the formation of crystalline sodium silicate in the matrix was more favorable to obtain higher strength. All of their studies all indicated that the lower modulus of the sodium silicate solution was more favorable to obtain higher strength. The reason for this analysis may be related, on the one hand, to the bilayer structure of the waterglass particle (Figure 7) [36], where a change in the modulus causes a change in the bilayer. The bilayer structure of the waterglass particle is an amorphous (SiO_2_)_m_ as the cores, the outer layer of the cores adsorbs a large amount of negatively valent H_3_SiO_4_^−^, the negatively valent H_3_SiO_4_^−^ and part of the Na^+^ are distributed within the dense layer, and another part of Na^+^ is free in the diffusion layer, which finally forms the bilayer. An increase in the modulus generates more negatively valent H_3_SiO_4_^−^, and more Na^+^ is adsorbed within the dense layer leading to a forced thinning of the diffusion layer and a decrease in the free Na^+^ and OH^-^ content. As per the “depolymerization-condensation” theory of Davidovits [37], the Si^4+^, Al^3+^, and Ca^2+^, which are involved in the formation of structure in raw materials, need to be activated by Na^+^ and OH^-^. Consequently, the high modulus leads to the reduction in Si^4+^, Al^3+^, and Ca^2+^ in the system, which cannot form a more dense and complete condensation structure, so the resistance to seawater immersion is poor. On the other hand, it is likely that the increment in WGM resulted in a decline in the relative amount of Na_2_O and a reduction in the impact of alkali activation, which led to a loss of strength [34]. It was more likely that this decrease in strength was connected to the bilayer architecture of the waterglass particle because the experimental design fixed the AE of group M at 9% (Table 4), i.e., the content of Na_2_O was the same.

### 3.3. X-Ray Diffraction (XRD) Analysis

An XRD diffraction study on GPM specimens with various AE, WGM, and immersion times was carried out in order to further investigate the internal process of the change in geopolymer material properties; the findings are shown in Figure 5. As demonstrated in Figure 5, the primary crystalline phase of GPM was quartz (PDF#33-1161), which is mostly produced from the geopolymer’s raw material and non-reactive material [38]. Comparing the XRD patterns under different AE and WGM in the figure, it is easy to see that the strength of the quartz phase diffraction peaks tended to decline with increasing AE and decreasing WGM, indicating that the unreacted material in GPM was reduced and the reaction degree was intensified, which is consistent with the change trend of compressive strength in the previous Section 3.2. It is further shown that AE and WGM primarily modified the seawater immersion resistance of GPM by affecting the raw materials’ degree of reactivity. A limited number of inconspicuous X-diffraction peaks appeared as calcite (PDF#05-0586) after 270 days of seawater immersion. Calcite itself has low strength, so it also contributes to the reduction in strength of the geopolymer material. No significant new crystalline phases appeared throughout the immersion cycle, and the loss of strength came mainly from changes in physical aspects.

### 3.4. Scanning Electron Microscope (SEM) Analysis

GPM specimens with various AE, WGM, and immersion times were chosen for SEM to explore the microstructure in order to further analyze the change in geopolymer material properties and the internal process of seawater immersion, and the findings are displayed in Figure 2. Figure 2a shows the scanned structure of GPM with AE at 3%, and it is clear from this that the GPM microstructure was poorly formed, with an overall loose and porous shape. A substantial amount of unreacted fly ash, pores, and a minor amount of gel state hydration products could be seen in the structure after being magnified by 2000 times. This is because Na^+^ helps to produce zeolite phase crystallization while OH^-^ in NaOH speeds up the dissolution of Si^4+^ and Al^3+^ in the raw material. Due to the low NaOH content in the low-AE environment, neither the OH^-^ nor the Na^+^ levels were sufficient to fully dissolve the Si^4+^ and Al^3+^ in the raw material, which prevents the network from fully polymerizing [28,39]. All of these factors might result in a lack of raw material reaction and a low concentration of gel structure created by condensation, which prevents the particles from binding together to form a compact structure. When the AE was raised to 9%, as shown in Figure 2b, the overall structure of GPM was more compact and the sample’s pores greatly diminished. It can be seen that the raw materials not involved in the reaction were significantly reduced after being magnified 3 k times, and instead, a large amount of geopolymer gel phase was formed to wrap some unreacted silica fume particles, which indicated that the increase in AE caused more raw materials to be dissolved and condensed into gel material to achieve the strength improvement. Figure 2c shows the scan structure when the modulus of GPM increased to 1.8. The internal structure of the geopolymer gel phase was less dense than the modulus of 1.2 (Figure 2b), which further explains that the increase in WGM causes a decrease in the resistance of GPM to seawater immersion, confirming the validity of the waterglass particle’s bilayer theory. Additionally, the presence of numerous microcracks observed with the electron microscope, combined with the XRD pattern analysis, may be ascribed to the internal dolomite reacting with alkaline materials to produce Mg(OH)_2_ with swelling properties, which weakens the bond between the geopolymer gel and the aggregate. The SEM picture of GPM following 270 days of seawater immersion is shown in Figure 2d. In comparison to the specimen without seawater immersion, two geopolymer gel phases with varying degrees of densification, i.e., the two areas A and B, were also noticed in addition to some unreacted raw components. Area B was close to the surface of the specimen and had a large number of light-colored sea salt crystals attached, which were absent from area A. This indicates that the GPM specimen had a good resistance to seawater invasion and seawater did not entirely infiltrate the interior. Additionally, the structural state of the area B was discovered to be irregularly fishnet-like when magnified 20,000 times, as shown in Figure 2d, and the structure was less dense in comparison to area A. It indicates that seawater immersion hydrolyzed part of the gel phase structure of GPM, which had a certain destructive effect on the structural denseness.

### 3.5. Predicted Results and Discussion of SVR Model

Utilizing the developed SVR model, the regression prediction of GPM strength under seawater immersion was carried out. Figure 8 displays the measured and predicted results of the acquired training and test sets.

The *MSE* and *R*^2^ results for the training and test sets are presented in Table 10. The training and test sets of sample A both had a low *MSE*, reaching 10^−3^, and *R*^2^ was above 0.97; the *MSE* of sample M also reached 10^−3^, and the *R*^2^ was more than 0.99. It is clear that the prediction results of both sample sets were in good agreement with measured results, and the SVR model was able to predict the time-varying UCS of GPM under seawater immersion with a good accuracy. By comparing the *MSE* and *R*^2^ of the two groups, it is obvious that the training effect for sample A was weak and the *R*^2^ of the training set was quite low, which was caused by the discrete character of the data. In contrast to WGM, the influence of *AE* variations on GPM strength had a wider range. The extreme difference of sample A was 39.42 MPa, whereas the extreme difference of sample M was 26.35 MPa. The dispersion of Sample A was larger than that of M; hence, the training impact of the regression model was generally subpar.

### 3.6. SVR Model Analysis

#### 3.6.1. Parameter Analysis

In this study, the RBF was used to train the SVR, and the optimal parameter penalty factor *C* and kernel function variance *g* of the model were obtained with the help of the cross-validation method. The results are illustrated in Table 11.

The penalty factor *C* represents the weight given to outlier samples, and the higher the value of *C*, the more samples must be used as a support vector for the model. The penalty factor *C* of sample A in this study was significantly larger than that of sample M, indicating that sample A required more support vectors for training. The influence distance of a single training sample was managed by the kernel function variance *g*. The radius of effect was narrower with a greater value of *g*, and the sample values must be highly similar to one another before they can be categorized as belonging to the same sort, which leads to the overfitting phenomena. Sample A’s *g* was modest, suggesting that the model support vector had a larger radius of sample effect during training. This phenomenon was still caused by the discrete nature of the data.

#### 3.6.2. Comparative Performance Evaluation

The same data were subjected to regression analysis using a neural network prediction model in order to compare and evaluate the applicability of the SVR model. Figure 9 shows the results of the regression analysis of GPM strength using the back propagation (BP) neural network prediction model for samples A and M after 270 days of seawater immersion.

The *R*^2^ of the BP neural network’s prediction results was calculated as 0.99 and 0.91, respectively. The *R*^2^ and goodness-of-fit of the BP neural network model decreased as compared to the SVR model (Table 10). It is worth noting that the prediction accuracy of the individual tests (Nos. 46 and 47) in Figure 9b significantly decreased, and the maximum prediction error reached 4.24 MPa, indicating the relatively low accuracy of the BP neural network model for predicting GPM strength under seawater immersion.

An attempt was made to further anticipate the compressive strength of GPM after 300 days of seawater immersion in order to confirm the prediction time range of the constructed SVR model, and the expected outcomes are shown in Figure 10.

The prediction accuracy of both models declined after 300 days of seawater immersion (No. 51–55) in comparison to 270 days, and the projected values of compressive strength were higher than the true values (Figure 10). This is due to the fact that when seawater immersion deepens, the UCS decay rate of GPM increases compared with the previous period, which causes a drop in forecast accuracy. When comparing the prediction accuracy of the two models, it is clear that the SVR model was relatively better, with a maximum error of 1.48 MPa in prediction (No. 53), whereas the highest error of the BP neural network reached 4.59 MPa (No. 55). It demonstrates that the BP neural network model had a very constrained prediction range capacity, whereas the SVR prediction model can essentially achieve the UCS prediction of GPM within 1–2 months.

In summary, the SVR model outperforms the artificial neural network model for predicting the strength of GPM under seawater immersion, and in the case of small samples and short-term prediction, the SVR model can more accurately predict the trend of the strength of geopolymer structures under seawater immersion, particularly the relationship with AE and WGM. Additionally, the created SVR model can forecast the compressive strength of GPM with high accuracy in 1–2 months, which can offer a reliable, scientific way for predicting the strength of geopolymer materials in actual offshore construction projects.

## 4. Conclusions

The effects of AE and WGM on the performance of GPM before and after seawater immersion were investigated, as well as the effective prediction of GPM strength under seawater immersion using the SVR prediction model. The following conclusions can be reached after discussing the results of the above sections:

(1) The time-varying law of UCS of GPM with seawater immersion was obtained. The UCS of GPM showed a trend of rapid increase followed by slow decrease. The rapid increment in strength from 0 to 60 days was related to the charge balancing of metal cations in seawater entering the reticular structure of the GPM; the gradual decline in strength after 60 days was attributed to the reduction in the gel phase structure.

(2) The influence of AE and WGM on the resistance of GPM to seawater immersion was studied. The GPM became less resistant to seawater immersion as AE increased because it produced more free alkali. The GPM construction revealed that the best resistance to seawater immersion when AE was in the range of 9%-12%. The diffusion layer of the bilayer structure inside the waterglass micelle attenuated as WGM increased (>1.0), and the resistance of GPM to seawater immersion decreased as a result.

(3) The mechanism of seawater immersion was exposed. The XRD and SEM data indicated that the hydrolysis of the gel phase in the geopolymer structure was primarily responsible for the immersion of seawater and did not produce a significant amount of new destructive material.

(4) Time-varying intensity regression and prediction of GPM were achieved. The SVR prediction model could more accurately estimate the trend of the UCS of GPM during seawater immersion, especially in respect of AE and WGM, with limited samples and short-term forecast. It could accurately forecast the UCS of GPM within 1–2 months, which can offer a reliable, scientific way to predict the strength of geopolymer materials in real offshore construction projects.

This work investigated the effects of AE and WGM on GPM performance before and after seawater immersion, and used the SVR model for the first time in predicting GPM strength under seawater immersion. However, the study still has some limitations, such as more complex influencing factors. Multi-influencing factors such as activator type and aggregate type can be introduced in future studies. Studying more influencing factors requires further improvement of the algorithm, so swarm intelligence algorithms such as the dragonfly algorithm (DA) and whale optimization algorithm (WOA) can be used in future studies [40,41,42,43].

## Figures and Tables

**Figure 1 materials-16-01244-f001:**
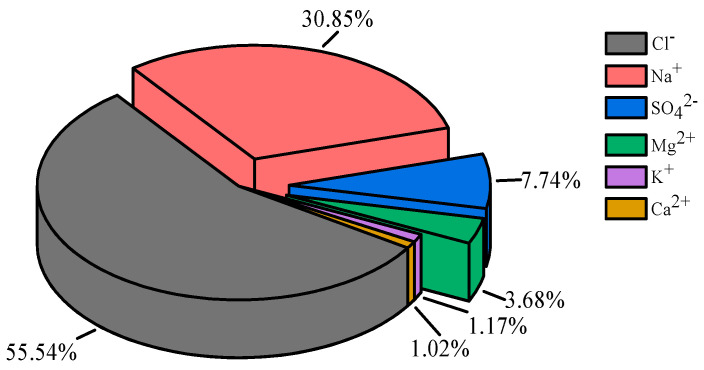
Mass percentage of main dissolved ions in artificial seawater.

**Figure 2 materials-16-01244-f002:**
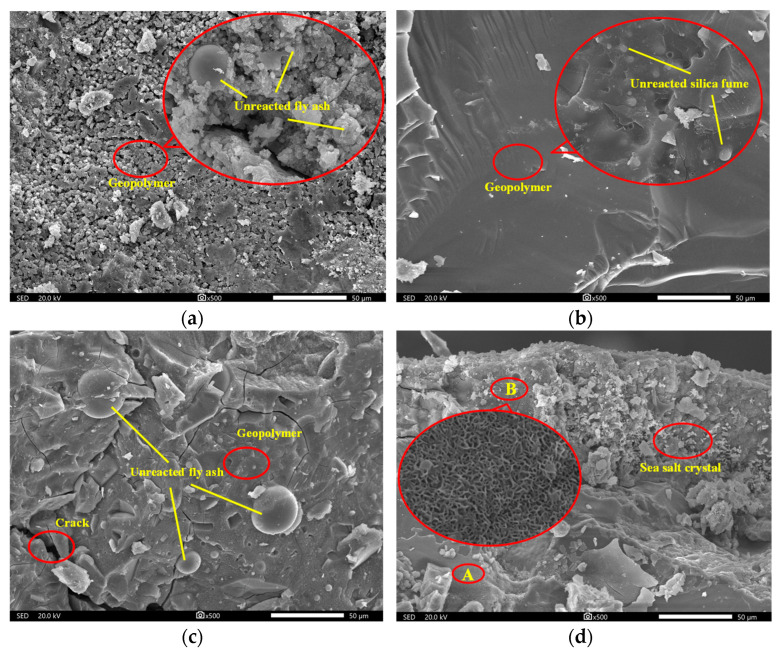
SEM images of GPM: (**a**) *Ae* = 3%, *Mo* = 1.2; (**b**) *Ae* = 9%, *Mo* = 1.2; (**c**) *Ae* = 9%, *Mo* = 1.8; (**d**) *Ae* = 9%, *Mo* = 1.8, eroded by 270 days.

**Figure 3 materials-16-01244-f003:**
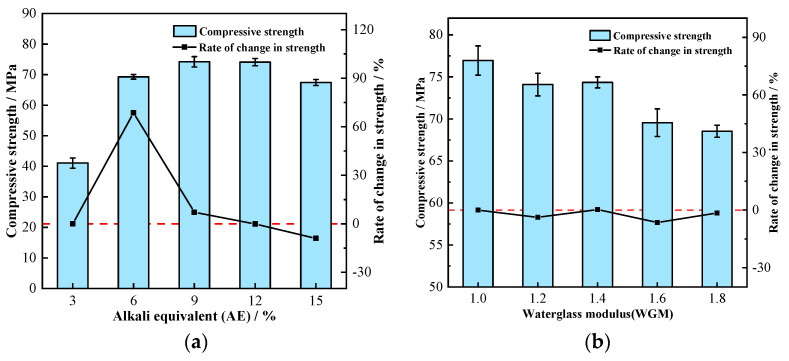
Relationship between initial compressive strength and activator parameters: (**a**) Relationship with AE; (**b**) Relationship with WGM.

**Figure 4 materials-16-01244-f004:**
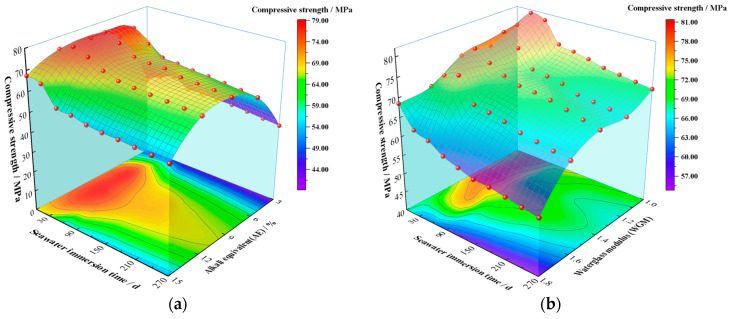
Effect of seawater immersion on the UCS of GPM: (**a**) Effect of seawater immersion time and AE; (**b**) Effect of seawater immersion time and WGM.

**Figure 5 materials-16-01244-f005:**
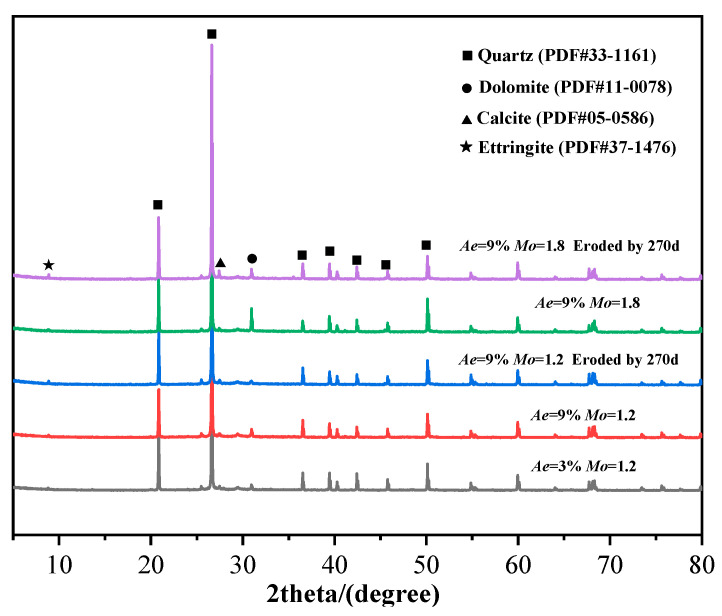
XRD patterns of GPM specimens.

**Figure 6 materials-16-01244-f006:**
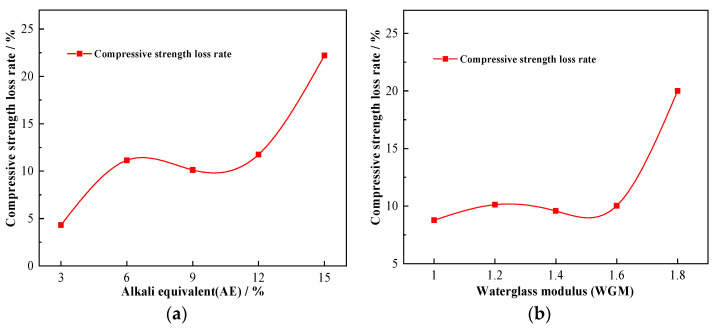
Compressive loss rate of GPM after 270 d of seawater immersion: (**a**) The impact of AE; (**b**) The impact of WGM.

**Figure 7 materials-16-01244-f007:**
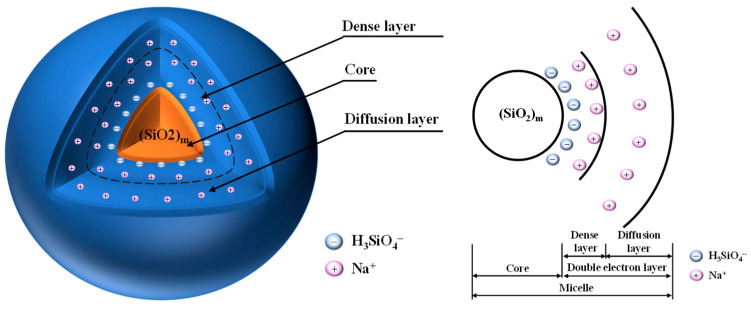
The bilayer structure of the waterglass particle.

**Figure 8 materials-16-01244-f008:**
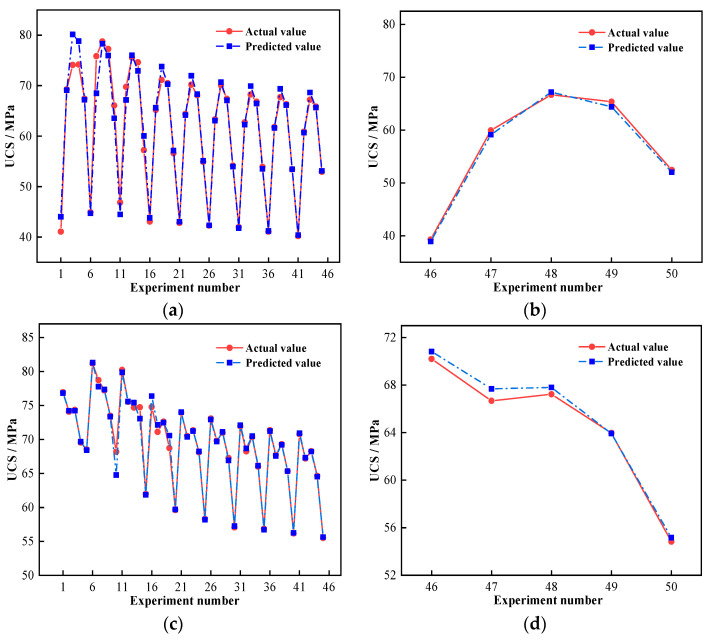
Comparison of prediction results of the training set and the test set: (**a**) Training set of sample A; (**b**) Testing set of sample A; (**c**) Training set of sample M; (**d**) Testing set of sample M.

**Figure 9 materials-16-01244-f009:**
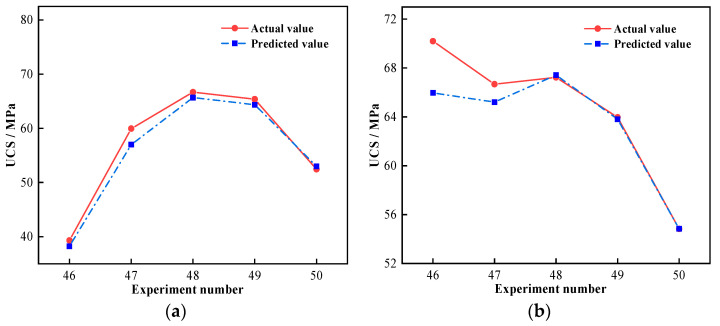
Comparison of prediction results of BP neural network: (**a**) Sample A; (**b**) Sample M.

**Figure 10 materials-16-01244-f010:**
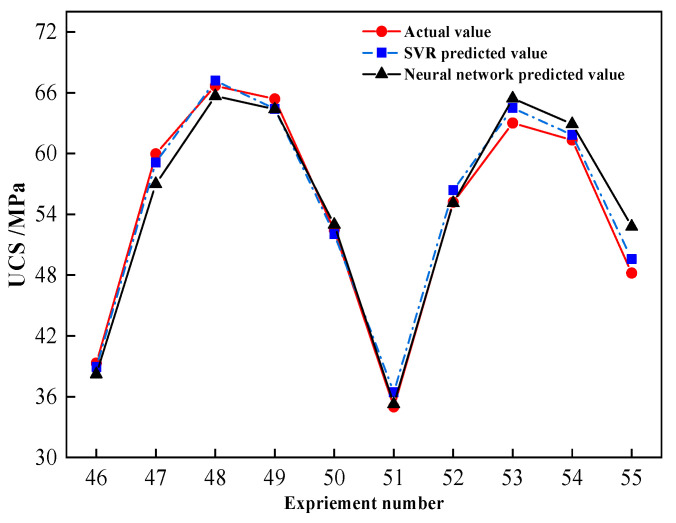
Comparison of predicted results of seawater immersion of sample A for 300 days.

**Table 1 materials-16-01244-t001:** Chemical composition of the binders.

Oxide Content	SiO_2_	Al_2_O_3_	Fe_2_O_3_	CaO	TiO_2_	SO_3_	K_2_O	MgO	Na_2_O
FA (wt %)	50.94	36.20	3.93	3.63	1.36	1.26	1.11	0.52	0.41
Slag (wt %)	32.73	14.61	0.27	36.50	0.68	2.45	0.43	7.87	0.28

**Table 2 materials-16-01244-t002:** Properties of the binders.

Materials	Density (g/cm^3^)	Specific Surface Area (m^2^/g)	Loss on Ignition /%
FA	2.42	0.43	1.51
Slag	3.00	0.60	2.30
SF	2.35	17.00	2.43

**Table 3 materials-16-01244-t003:** Mixture proportions of group A (kg/m^3^).

No.	*Ae*/%	FA	SF	Slag	Sand	NaOH	Waterglass	Water
A3	3	105	180	315	1200	14.90	75.06	209
A6	6					29.79	150.11	179
A9	9					44.69	225.17	148
A12	12					59.59	300.22	117
A15	15					74.49	375.28	86

**Table 4 materials-16-01244-t004:** Mixture proportions of group M (kg/m^3^).

No.	*Mo*	FA	SF	Slag	Sand	NaOH	Waterglass	Water
M10	1.0	105	180	315	1200	48.86	187.64	168
M12	1.2					44.69	225.17	148
M14	1.4					40.53	262.69	128
M16	1.6					36.36	300.22	108
M18	1.8					32.20	337.75	88

**Table 5 materials-16-01244-t005:** Chemical Composition of artificial seawater, (g/L).

Compound	Standard Concentration	Artificial Seawater Concentration
NaCl	24.53	73.59
MgCl_2_·6H_2_O	11.11	33.33
Na_2_SO_4_	4.09	12.27
CaCl_2_	1.16	3.48
KCl	0.695	2.09
NaHCO_3_	0.201	0.60

**Table 6 materials-16-01244-t006:** Data for sample A.

Experiment No.	Influencing Factors(*x*_i_)		Measured Compressive Strength (*y*_i_)/MPa
	Immersion Time/d	*Ae*/%	
1	0	3	41.08
2	0	6	69.27
3	0	9	74.19
4	0	12	74.09
5	0	15	67.41
6	30	3	44.90
7	30	6	75.83
…	…	…	…
48	270	9	66.68
49	270	12	65.38
50	270	15	52.44

**Table 7 materials-16-01244-t007:** Data for sample B.

Experiment No.	Influencing Factors(*x*_i_)		Measured Compressive Strength (*y*_i_)/MPa
	Immersion Time/d	*Mo*	
1	0	1	76.95
2	0	1.2	74.19
3	0	1.4	74.35
4	0	1.6	69.55
5	0	1.8	68.54
6	30	1	81.18
7	30	1.2	78.73
…	…	…	…
48	270	1.4	67.23
49	270	1.6	62.57
50	270	1.8	54.83

**Table 8 materials-16-01244-t008:** Mass change of Group A.

No.	Indicators	Seawater Immersion Time/d					
		0	30	60	120	180	270
A3	Mass/g	754.22	763.52	765.67	769.50	760.10	762.80
	Loss rate/%	-	1.23	0.28	0.50	−1.22	0.36
A6	Mass/g	744.92	759.67	760.07	764.40	760.30	759.80
	Loss rate/%	-	1.98	0.05	0.57	−0.54	−0.07
A9	Mass/g	746.20	750.45	768.42	769.70	762.60	763.40
	Loss rate/%	-	0.57	2.39	0.17	−0.92	0.10
A12	Mass/g	735.37	738.42	757.55	756.70	756.10	748.80
	Loss rate/%	-	0.41	2.59	−0.11	−0.08	−0.97
A15	Mass/g	738.98	744.80	754.68	754.30	742.60	738.92
	Loss rate/%	-	0.79	1.33	−0.05	−1.55	−0.50

**Table 9 materials-16-01244-t009:** Mass change of Group M.

No.	Indicators	Seawater Immersion Time/d					
		0	30	60	90	180	270
M10	Mass/g	755.37	757.38	761.20	763.80	758.60	761.20
	Loss rate/%	-	0.27	0.50	0.34	−0.68	0.34
M12	Mass/g	746.20	750.45	768.42	769.70	762.60	763.40
	Loss rate/%	-	0.57	2.39	0.17	−0.92	0.10
M14	Mass/g	751.67	760.28	767.67	760.80	767.20	760.80
	Loss rate/%	-	1.15	0.97	−0.89	0.84	−0.83
M16	Mass/g	743.35	754.28	760.47	755.80	760.50	761.70
	Loss rate/%	-	1.47	0.82	−0.61	0.62	0.16
M18	Mass/g	733.93	749.40	746.65	740.60	738.40	733.80
	Loss rate/%	-	2.11	−0.37	−0.81	−0.30	−0.62

**Table 10 materials-16-01244-t010:** *MSE* and *R*^2^ of the training and testing sets.

Sample	Training Set		Testing Set	
	*MSE*	*R* ^2^	*MSE*	*R* ^2^
A	0.00103	0.9709	0.00118	0.9974
M	0.00200	0.9919	0.00224	0.9966

**Table 11 materials-16-01244-t011:** The best parameters *C* and *g*.

Sample	Penalty Factor *C*	Kernel Function Variance *g*
A	1024	0.0884
M	90.5097	1

## Data Availability

Not applicable.

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
