# Peer review of "Time-Varying Pattern and Prediction Model for Geopolymer Mortar Performance under Seawater Immersion"

_materials, 2023, doi:10.3390/ma16031244_

Round 1

Reviewer 1 Report

This study presents a detailed analysis of geopolymer mortar erosion under the influence of artificial seawater, does not contain plagiarism, and looks at the problem from all the important aspects. However, there are significant concerns, as follows.

-    Chapter 2 seems unnecessary because the manuscript itself is long enough, and the process of geopolymerization has already been described many times in the literature.
-    Explain in more detail which slag was used to produce the mortars.
-    Why were not mixture proportions used as influencing indicators in mathematical modeling? Those are also very important factors.
-    I believe that the term “time varying” is a bit too prominent, since only the fresh and 28-days old samples are tested (The main title and a sub-chapter title).
-    Is it possible that the XRD peak at about 9 2-Theta could be of ettringite? Is there any mica in fly ash or slag?

Reviewer 2 Report

The manuscript entitled ‘Time-varying pattern and prediction model for geopolymer mortar performance under seawater erosion” is in line with the Materials journal. This article is based on original research. The topic is up-to-date and interesting to potential readers The manuscript is well composed; however, it requires minor changes before publication, such as:

·        All article: use the template given on the publisher site.

·        Introduction: new literature should be used for the analysis, the authors based mainly on the old literature.

·        Introduction (last paragraph): stress the literature gap and novelty aspects presented in the article.

·        Chapter 2.: it is a theoretical part; it should be removed or joined with introduction part.

·        Chapter 3.1. add some information about the material origin.

·        Chapter 3.3: water absorption test should be characterized more carefully (named by the authors as a changing weight).

·        Figure 2. please define ‘%’ by mass or other value?

·        Chapter 3.: the numerical methods should be characterized in this part.

Reviewer 3 Report

Find from the attached document.

Round 2

Reviewer 1 Report

The paper is now suitable for publication and has been sufficiently improved.

Author Response

Thank you for your comments concerning our manuscript entitled “Time-varying pattern and prediction model for geopolymer mortar performance under seawater erosion” (ID: materials-2111401). We would like to thank the reviewer for his thoughtful and productive comments. 

Reviewer 3 Report

All comments have been addressed.

Author Response

Thank you for your comments concerning our manuscript entitled “Time-varying pattern and prediction model for geopolymer mortar performance under seawater erosion” (ID: materials-2111401). We would like to thank the reviewer for your thoughtful and productive comments.